# Characterization of Nickel Oxide Nanoparticles Synthesized under Low Temperature

**DOI:** 10.3390/mi12101168

**Published:** 2021-09-28

**Authors:** Sung-Jei Hong, Hyuk-Jun Mun, Byeong-Jun Kim, Young-Sung Kim

**Affiliations:** 1Korea Electronics Technology Institute, Seongnam 13509, Korea; hengel20@naver.com; 2Nano IT Convergence Engineering, Graduate School of NIDE Fusing Technology, Seoul National University of Science & Technology, Seoul 01811, Korea; kbj1020@seoultech.ac.kr

**Keywords:** nickel oxide nanoparticles (NiO NPs), wet chemical synthesis, heat-treatment under low temperature, hexagonal structure, FT-IR spectra, Ni-O bond, UV-Vis spectra, blue shift, quantum confinement effect

## Abstract

In this study, ultrafine nickel oxide nanoparticles (NiO NPs) were well synthesized using a simple wet chemical method under low temperature, 300 °C. An Ni(OH)_2_ precursor was well precipitated by dropping NH_4_OH into an Ni(Ac)_2_ solution. TG-DTA showed that the weight of the precipitate decreases until 300 °C; therefore, the precursor was heat-treated at 300 °C. X-ray diffraction (XRD) patterns indicated that hexagonal-structured NiO NPs with (200) preferred orientation was synthesized. In addition, BET specific surface area (SSA) and HRTEM analyses revealed that spherical NiO NPs were formed with SSA and particle size of 60.14 m^2^/g and ca. 5–15 nm by using the low temperature method. FT-IR spectra of the NiO NPs showed only a sharp vibrating absorption peak at around 550 cm^−1^ owing to the Ni-O bond. Additionally, in UV-vis absorption spectra, the wavelength for absorption edge and energy band gap of the ultrafine NiO NPs was 290 nm and 3.44 eV.

## 1. Introduction

Nickel oxide (NiO) is an important transition metal oxide material with a cubic lattice structure used in numerous applications [1,2]. NiO nanoparticles (NPs) have a wide range of applications such as battery electrodes, photo-electron devices, ion storage materials, gas sensors, magnetic materials, thermoelectric materials, catalysts, fuel cells, dye-sensitized photocathodes, electrochromic films, anticancer properties, cytotoxic activity and non-enzymatic glucose sensors, etc. [3,4]. In addition, NiO nanoparticles (NPs) are studied because nano-dimensional particles differ from the majority of nanoparticles in a number of properties, including surface area to volume ratio and electro-optical, magneto-optical, chemical and mechanical properties, which lead to unique optical, electronic and physiochemical properties [5]. With advancements in all areas of industry and technology, interest has been focused on nanoscale materials, stemming from the fact that new properties are required at this length scale and, equally important, that these properties change with their size and morphology [6]. Size reduction affects most physical properties (structural, magnetic, optical, dielectric, thermal, etc.) due to surface effects and quantum size effects [7].

Several methods have been attempted to prepare NiO NPs [8,9,10,11,12], but many come with disadvantages because the process is complicated and consumes large amounts of energy to induce a large amount of carbon. Although some methods have been reported as synthesizing them while decreasing consumption energy [8], there are still issues, including the limitation of manufacturing capacity for use in production lines, etc. Accordingly, simple and cost-effective synthetic methods are important in order to reach global carbon neutrality.

In this study, we synthesized NiO NPs by using an eco-friendly method under a low temperature that uses less energy. Although we selected a simple wet chemical method that is generally used in a production line, we synthesized NiO NPs by using acetate-based raw materials that do not contain harmful substances such as Cl^-^ or NO_3_^-^, resulting in a decrease of wastewater. Existing synthetic methods using materials containing harmful substances require several washing steps with water to remove those substances, and a significant amount of wastewater is generated during the repeated cleaning steps, whereas the acetate substance is easily washed into water, with the result that the number of washing steps and the amount of wastewater generation can be decreased by applying acetate-based raw materials. In addition, we attempted to lower the heat-treatment temperature to optimize ultrafine-sized NiO NPs as well as to reduce carbon emission.

## 2. Materials and Methods

### 2.1. Preparation of NiO NPs

Nickel (II) acetate tetrahydrate (Ni(CH_3_CO_2_)_2_·4H_2_O) was used as a raw material. Salt was dissolved into ethanol, and 2.0 M ammonia (NH_4_OH) was added dropwise into the solution to precipitate a precursor of NiO NPs. The precipitate was then centrifugated and dried at 70 °C. To determine the lowest heat-treatment temperature where components are thermally decomposed, except Ni to be oxidized, examination of thermal behavior of the precipitate was conducted using thermogravimetric analysis (TGA) and differential thermal analysis (DTA) at temperatures ranging from 25 °C to 550 °C with a rising rate of 10 °C/min. Based on the result, heat-treatment temperature was determined from the thermally decomposed point, 300 °C, to 600 °C at an increment of 100 °C, and four NiO NPs samples were prepared.

### 2.2. Characterizations of NiO NPs

The prepared NiO NPs samples were characterized as follows: crystal structures were analyzed by using an X-ray diffractometer (XRD, Empyrean, Malvern Panalytical B.V., Almelo, Netherlands) with Cu K radiation over a range of 2θ angles from 20° to 80°. In addition, XRD pattern of the precipitate before heat-treatment was analyzed to determine its compound forms. Their surface areas were measured with a Brunauer, Emmett & Teller surface area analyzer (BET SSA, ASAP2020, Micromeritics, Norcross, GA, USA). Their particle size and morphology were observed using a high-resolution transmission electron microscope (HRTEM, JEM-2010, JEOL, Tokyo, Japan). Their chemical structures were examined using Fourier transform infrared spectroscopy (FT-IR, IRAffinity-1S, Shimadzu, Kyoto, Japan) in the range of 400–4000 cm^−^^1^. In addition, their UV-Vis absorption spectra were investigated at room temperature by using a UV-Vis spectrophotometer (CM-3600d, Konica Minolta, Tokyo, Japan) in the range of 300–800 nm.

## 3. Results

### 3.1. Crystal Structure

Firstly, we analyzed the crystal structure of the precipitate from the Ni (II) acetate reacted with ammonia solution. Its XRD pattern is shown in Figure 1. Most peaks are indexed to those of a pure phase of Ni(OH)_2_ with a hexagonal structure [13], i.e., it appears to be reduced and precipitated in the form of Ni(OH)_2_ as follows:Ni(CH_3_CO_2_)_2_ + 2NH_4_OH → Ni(OH)_2_ + 2NH_4_(CH_3_CO_2_)(1)

Next, we analyzed the thermal behavior of the Ni(OH)_2_ precipitate. In Figure 2, the thermal weight of the precipitate (red color) tends to decrease as the temperature increases, and the weight rapidly decreases from 250 °C to 300 °C, maintaining a constant of about 55% weight above 300 °C. In addition, from the DTA graph of the Ni(OH)_2_ precipitate (blue color), an exothermic behavior was observed in the range of 250 °C to 300 °C. The decrease in weight is due to the decomposition and evaporation of H and O components, excepting the Ni component, which were included in the Ni(OH)_2_ precipitate. Therefore, it appears possible to prepare crystalline NiO NPs by a heat-treatment of above 300 °C.

The XRD patterns of the four NiO NPs samples are presented in Figure 3a–d. Despite the differences in heat-treatment temperature of each sample, all samples show a similar diffraction pattern. The NiO NPs heat-treated at 300 °C in Figure 3a have main diffraction peaks at 37.23°, 43.32°, 62.80°, 75.47° and 79.47°, indicating <111>, <200>, <220>, <311> and <222> directions, respectively. Among them, the growth was greatest in the <200> direction, indicating a hexagonal structure with the <200> preferred orientation. Additionally, in Figure 3b–d, the samples heat-treated at 400 °C, 500 °C, and 600 °C showed diffraction patterns similar to those of 300 °C. Thus, it is evident that all the NiO NPs samples have the same hexagonal structure.

### 3.2. Particle Size

From the XRD peaks, the full-width-half-maximum (FWHM) of the <200> peak decreased as the heat-treatment temperature increased. This result indicates that the size of the NiO NPs heat-treated at lower temperatures is ultrafine nanocrystal according to the Scherrer’s Equation [14]. From the XRD peaks, particle size can be calculated by using Scherrer’s Equation as
t = 0.9 λ/B cos θ_B_(2)
where t, λ, B and θ_B_ are particle size, wavelength (0.1542 nm for CuK_α_ radiation), full-width-half-maximum (FWHM) of a peak in radians and diffracted angle, respectively. In Equation (2), the peak intensity increases along with a reduction in the peak half width, indicating the growth of NiO NPs. Thus, as the FWHM is widened, the particle size is reduced. In Figure 4, calculated with Equation (2), particle size of NiO NPs heat-treated at 300 °C, 400 °C, 500 °C and 600 °C was 10.1 nm, 17.0 nm, 24.6 nm and 33.6 nm, respectively. Particle size increased in a linear proportion to the heat-treatment temperature.

BET SSA of the samples were measured in the next stage; the results are shown in Figure 5. The SSA of the NiO NPs heat-treated at 300 °C is above 60.14 m^2^/g. Furthermore, in the cases of samples heat-treated at higher temperatures of 400 °C, 500 °C and 600 °C, SSA decreased to 27.19 m^2^/g, 15.67 m^2^/g and 9.94 m^2^/g, respectively. The SSA heat-treated at 600 °C decreased to roughly one sixth of that heat-treated at 300 °C.

Particle size was calculated from the BET SSA as follows (see Equation (3)):D = 6/ρ·d(3)
where D, ρ and d are particle size, SSA and density, respectively. In Figure 5, the calculated particle size of NiO-NPs heat-treated at 300 °C, 400 °C, 500 °C and 600 °C was 15.0 nm, 33.1 nm, 57.4 nm and 90.5 nm, respectively. The SSA tends to decrease in inverse proportion to the heat-treatment temperature. In addition, in the conversion to particle size, the particle size increases in proportion to the heat-treatment temperature. It was found that there is a difference in the conversion of particle size values between XRD FWHM and BET SSA. It has been reported previously [15] that these differences are owing to measurement principles. In case of XRD FWHM, an X-ray can penetrate through the crystal size to provide information; therefore, the calculation of size is not related to the particles but to the crystals. In addition, it is hard to provide separation and distinguish between broadening through the crystallite size and broadening due to other parameters and factors, whereas in the case of BET SSA, under normal atmospheric pressure and at the boiling temperature of liquid nitrogen, the amount of nitrogen absorbed in relation to pressure gives the SSA of the nanoparticles. Errors may always occur, but successful calculation methods are those that decrease the errors in the best possible way to yield more accurate data. Although there is such a difference here, the results show that the increase in particle size due to the increase in heat-treatment temperature is consistent.

Thus, to certify the particle sizes of NiO NPs heat-treated at these temperatures, the samples were observed with HRTEM. In the case of NiO NPs heat-treated at 300 °C, shown in Figure 6a, ultrafine spherical particles with particle sizes of ca. 5~15 nm were observed. When the heat-treatment temperature increased to 400 °C, shown in Figure 6b, NiO NPs grew to ca. 20~40 nm, and particles were slightly agglomerated. The agglomerations between particles became more evident as heat-treatment temperature increased. In the case of NiO NPs heat-treated at 500 °C, shown in Figure 6c, particles of ca. 30~70 nm were observed. Additionally, particles of ca. 40~120 nm were observed when the heat-treatment temperature increased to 600 °C, as shown in Figure 6d.

Similar to the particle sizes calculated from XRD FWHM and BET SSA, it was clearly observed that particle size increased as the heat-treatment temperature increased. Also, it became evident that NiO NPs were more agglomerated as the heat-treatment temperature was raised. This result is attributed to the diffusion between particles activating as heat-treatment temperature increases. Details on this behavior are further reported in the discussion chapter.

### 3.3. FT-IR Spectra

To clarify chemical structures of Ni-O bonds in the NiO NPs, the characterization of the molecular structure and nature of chemical bonding in NiO NPs were assessed using FT-IR analysis. In Figure 7, the FT-IR spectra of NiO NPs are shown after heat-treatment at 300 °C, 400 °C, 500 °C and 600 °C, respectively. All samples showed a similar FT-IR vibration absorption pattern, and a sharp absorption peak was only observed around 550 cm^−1^ with a wavenumber (ν) indicating that it can be attributed to the oscillation of Ni-O [16]. The intensity of the absorption peak was also similar in the four samples. From these results, it is inferred that NiO NPs exhibit a constant vibration absorption peak at a constant wavenumber (ν) despite differences in particle size.

### 3.4. UV-Vis Absorption Spectra

UV-vis absorption behaviors of NiO NP samples were analyzed to gather information on its optical band gap. In Figure 8a, the UV-vis spectra of the NiO NPs are shown after heat treatment at 300 °C, 400 °C, 500 °C and 600 °C, respectively. The absorption tends to increase towards the shorter wavelength region, showing a phenomenon in which the absorption rate increases in the blue shift, i.e., UV region [17]. In addition, the wavelength for absorption edge of NiO NPs heat-treated at 300 °C, 400 °C, 500 °C and 600 °C is 290 nm, 301 nm, 317 nm and 338 nm, respectively. As the heat-treatment temperature is lowered, the wavelength decreases with it. Moreover, as the heat-treatment temperature is lowered, the absorbance at the absorption edge is raised. This decrease in particle size may be responsible for the shifting of absorption bands towards a lower wavelength [18].

In the next step, the energy band gap of the NiO NPs was calculated by a well-known relation given by Tauc, using Equation (4) [19]:(αhυ)^1/n^ = A(hυ − E_g_)(4)
where α, A, h, υ, n (=1/2 for direct band gap) and E_g_ are optical absorption coefficient, absorbance, Planck’s constant, frequency of light, constant related to mode of transition and band gap energy, respectively. Band gap energy was calculated by extrapolating the straight region of the graph plotted between “hυ” and (αhυ)^2^. The results show that the band gap energy of NiO NPs heat-treated at 300 °C, 400 °C, 500 °C and 600 °C is 3.44 V, 3.38 eV, 3.25 eV and 2.86 eV, respectively, as is shown in Figure 8b. As the heat-treatment temperature is lowered, the energy band gap is raised. This tendency appears to be related to differences in particle size; details on this tendency are discussed further in the next chapter.

## 4. Discussion

### 4.1. Particle Size

The variation in particle size during heat-treatment temperature change is attributed to particle growth driven by heat-treatment temperature. One mechanism used to increase nanoparticle size is due mainly to particle surface migration [20]. According to transformation kinetics,
D = exp(−Q/kT)(5)
where D, Q, k and T are mean particle size, activation energy for particle surface migration, Boltzmann constant and heat-treatment temperature, respectively. When the activation energy for particle growth becomes small at constant particle size, the surface activity of NiO NPs as a function of its temperature becomes high, and the growth of nanoparticles can be suppressed as the heat-treatment temperature is lowered. Thus, the lower temperature is necessary to synthesize smaller sized nanoparticles, and 300 °C is the lowest heat-treatment temperature for the crystalline NiO NPs.

### 4.2. FT-IR Spectra

In FT-IR spectra, the absorption peaks and their positions are attributed to the chemical composition, crystalline nature and morphology of the material, and the absorption peaks below 1000 cm^−1^ are considered to be essential for studying presence of Ni–O bonds in NiO NPs [17]. In Figure 7, the sharp vibration peaks at around 550 cm^−1^ in the spectra appear to relate to the Ni-O stretching [16]. In addition, any other vibration absorption patterns which appeared by peaks related to organic bonding, such as O-H, C-O, C-C, C=O and C=C, were scarcely observed, indicating that there is a pure Ni-O bond with few organic residues, i.e., the NiO NPs have high purity. Beyond this, it is known that NiO NPs exhibit a constant vibration absorption peak at a constant wavenumber (ν) regardless of differences in particle size. This is assumed to be related to the lattice parameter, i.e., from the XRD patterns, it is known that the four NiO NP samples have the same lattice parameter, 0.418 nm. Because of the same bonding length between Ni and O of all the samples, it is theorized that they exhibit almost the same vibration absorption peak at a constant wavenumber, however, further study is required to identify the relationship clearly.

### 4.3. UV-Vis Absorption Spectra

UV-visible absorption spectroscopy is an important method for estimating the energy structures and optical properties of nanoparticles. Discussion on the increased trend of band gap energy as it relates to decreases in particle size is well documented in existing literature [21,22]. It is additionally well documented that semiconductors with nanoscale size show a blue shift in their spectra due to quantum confinement effects [21]. The phenomenon of the blue shift occurs with an attendant increase in the band gap value, which is evidence of a quantum confinement effect. This implies that the band gap value became larger with decrease in particle size. Generally, as particle size decreases, the wavelength of the maximum exciton absorption (λ_max_) decreases due to the quantum confinement of the photo generated electron-hole carriers [22]. It is noted that NiO NPs exhibit a blue shift in the absorption onset; this blue shift in the absorption spectrum is mainly attributed to the confinement of charge carriers in the NiO NPs. α obeys Equation (4) for high photon energies (hν), and the fundamental absorption corresponding to the optical transition of the electrons from the valence band to the conduction band can be used to determine the nature and value of E_g_ of the NiO NPs.

## 5. Conclusions

In this study, ultrafine NiO NPs were well synthesized under low temperature, 300 °C, by using a simple wet chemical method with an eco-friendly Ni(Ac)_2_ solution. NiO NPs were well crystallized with hexagonal structures having <200> preferred orientation even at the lowest temperature. In addition, spherical NiO NPs were well formed with SSA and particle size of 60.14 m^2^/g and ca. 5–15 nm. FT-IR spectra of the NiO NPs showed a sharp vibrating absorption peak only at around 550 cm^−1^ owing to the Ni-O bond. UV–vis absorption spectra revealed that NiO NPs have the wavelength for absorption edge and their energy band gap is 290 nm and 3.44 eV, respectively. It is expected that the application of this eco-friendly method will contribute to the improvement of manufacturing environment and carbon neutrality.

## Figures and Tables

**Figure 1 micromachines-12-01168-f001:**
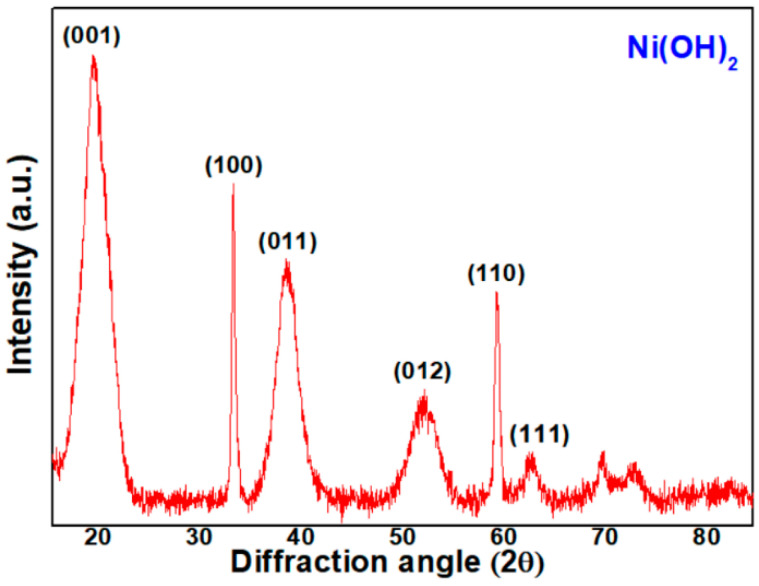
XRD pattern of the precipitate from Ni(II) acetate with ammonia.

**Figure 2 micromachines-12-01168-f002:**
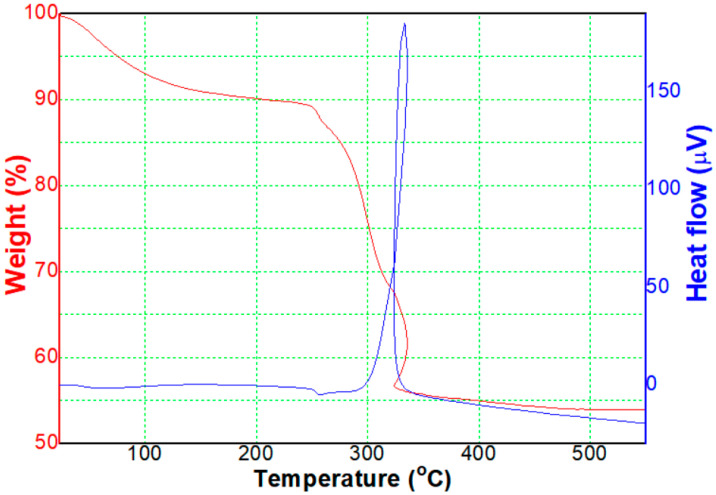
Thermal behavior of the precipitate.

**Figure 3 micromachines-12-01168-f003:**
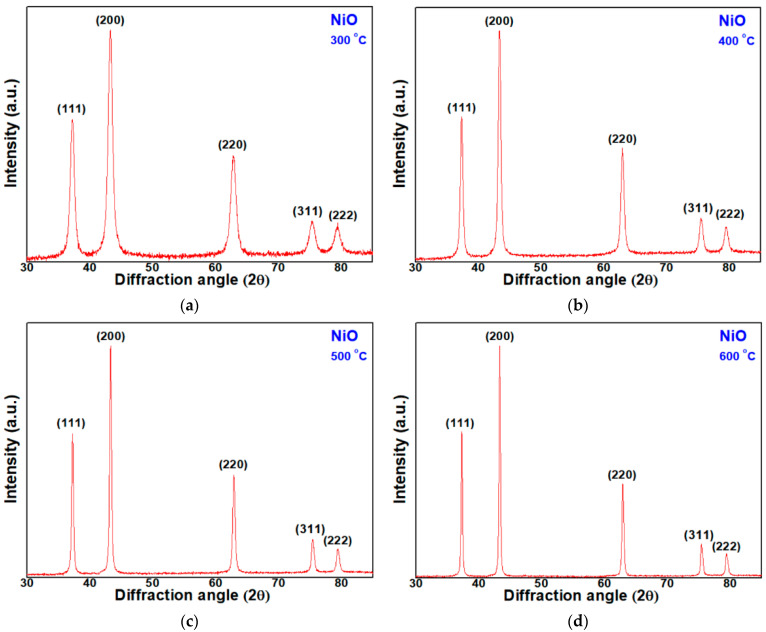
XRD patterns of NiO NPs heat-treated at (**a**) 300 °C; (**b**) 400 °C; (**c**) 500 °C; (**d**) 600 °C.

**Figure 4 micromachines-12-01168-f004:**
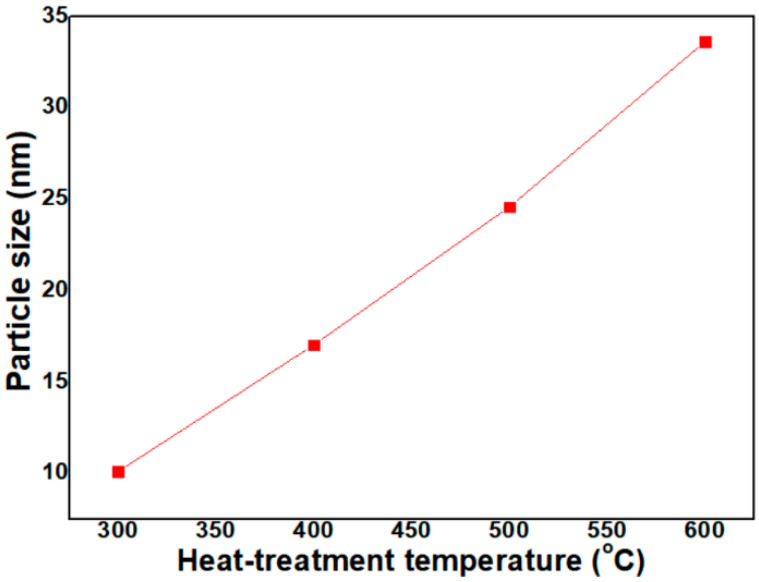
Variation of particle size of NiO NPs with heat-treatment temperature (calculated by using Equation (2)).

**Figure 5 micromachines-12-01168-f005:**
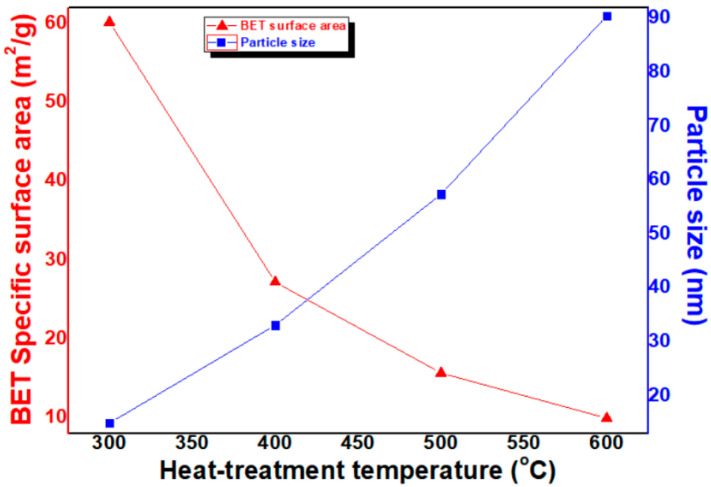
Variation of BET SSA and particle size (calculated by using Equation (3)) of NiO NPs with heat-treatment temperature.

**Figure 6 micromachines-12-01168-f006:**
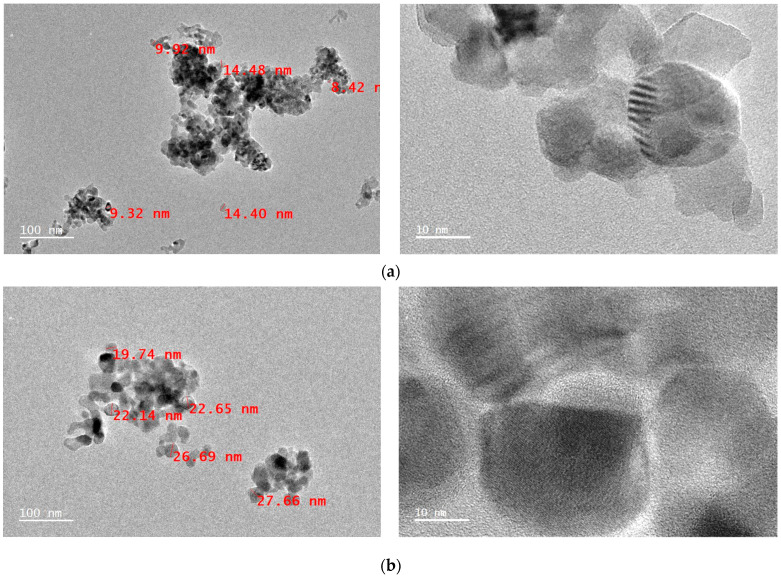
HRTEM observations of NiO NPs heat-treated at (**a**) 300 °C; (**b**) 400 °C; (**c**) 500 °C; (**d**) 600 °C.

**Figure 7 micromachines-12-01168-f007:**
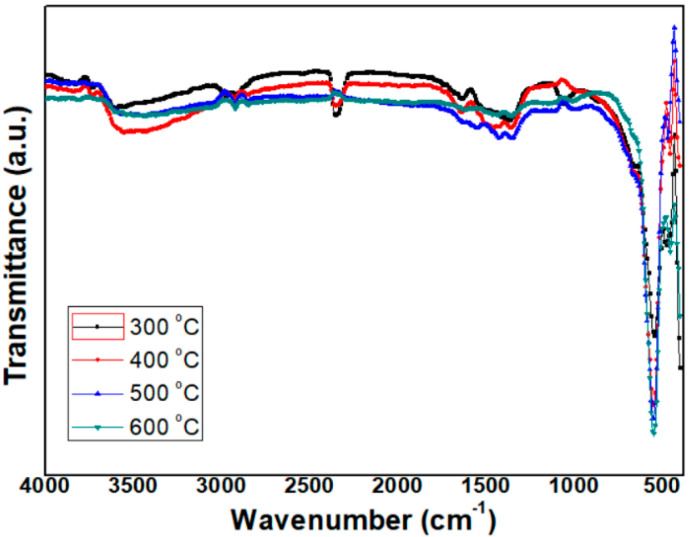
FT-IR spectra of NiO NPs heat-treated at various temperature.

**Figure 8 micromachines-12-01168-f008:**
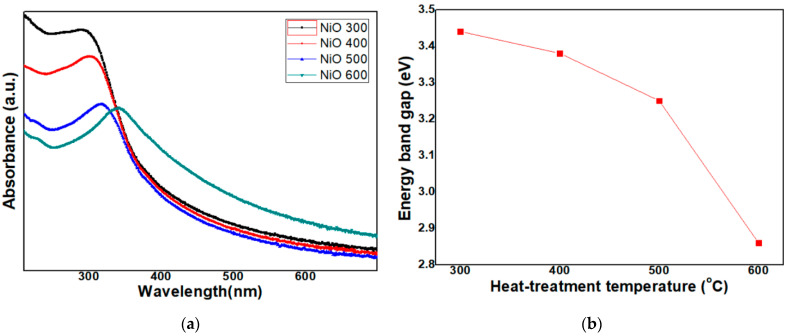
(**a**) UV-vis absorption spectra; (**b**) Energy band gap of NiO NPs heat-treated at 300 °C, 400 °C, 500 °C and 600 °C, respectively.

## Data Availability

Data are available upon request from the corresponding author.

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
