# Peer review of "Characterization of Nickel Oxide Nanoparticles Synthesized under Low Temperature"

_micromachines, 2021, doi:10.3390/mi12101168_

Round 1
Reviewer 1 Report
The manuscript can be published in Micromachines. However, the authors should provide an explanation about the numerical mismatch between the XRD and the BET SSA estimation of particle size.
Author Response
Dear Reviewer 1;
Thank you for your kind and valuable comments.
We prepared a response according to your comments.
Please find an attached file.
I would appreciate it if you could review it.
Thank you for your time and considerations.
Sincerely yours
Sung-Jei Hong
KETI

Reviewer 2 Report
The manuscript deal with the synthesis of Nickel Oxide nanoparticles under low temperatures methodology.
First of all, the intentions of the authors are clearly explained in the introduction section, supported by appropriate literature and are genuine and concretely demonstrated through the whole manuscript. The disadvantages in terms of sustainability and global carbon neutrality in the generally used thermal methods for NPs synthesis is of central importance nowadays.
The strategy followed by the authors is simple yet effective. The choice of the starting materials, the procedures and the characterization data and discussion are well assessed and precise.
The manuscript is suitable for publication after only some minor corrections as follows:
- in the reaction (1) for the formation of Ni precursor “NH4H” should be corrected with “NH4OH”
- In the introduction section could be explained more in details the choice of the acetate-based starting materials in order to avoid water consumption.
- The authors should also insert TEM (or HRTEM) imagines which allows to understand the particle size distribution which is not reported in the manuscript.
- In the conclusion section, the authors could be more incisive into highlight the importance of their work.
Author Response
Dear Reviewer 2;
Thank you for your kind and valuable comments.
We prepared a response according to your comments.
Please find an attached file.
I would appreciate it if you could review it.
Thank you for your time and considerations.
Sincerely yours
Sung-Jei Hong
KETI
